# Position: Editing Large Language Models Poses Serious Safety Risks

**Paul Youssef** [1]  **Zhixue Zhao** [2]  **Daniel Braun** [1]  **Jörg Schlötterer** [1][3]  **Christin Seifert** [1]

## Abstract

Large Language Models (LLMs) contain large amounts of facts about the world. These facts can become outdated over time, which has led to the development of knowledge editing methods (KEs) that can change specific facts in LLMs with limited side effects. This position paper argues that editing LLMs poses serious safety risks that have been largely overlooked. First, we note the fact that KEs are widely available, computationally inexpensive, highly performant, and stealthy makes them an attractive tool for malicious actors. Second, we discuss malicious use cases of KEs, showing how KEs can be easily adapted for a variety of malicious purposes. Third, we highlight vulnerabilities in the AI ecosystem that allow unrestricted uploading and downloading of updated models without verification. Fourth, we argue that a lack of social and institutional awareness exacerbates this risk, and discuss the implications for different stakeholders. We call on the community to (i) research tamper-resistant models and countermeasures against malicious model editing, and (ii) actively engage in securing the AI ecosystem.

## 1. Introduction

LLMs are utilized in a multitude of applications across various domains (Brahmavar et al., 2024; Van Veen et al., 2024). A primary factor contributing to the widespread popularity of LLMs is their capacity to function as repositories of knowledge, which can be effortlessly queried in natural language (Petroni et al., 2019; Roberts et al., 2020; Youssef et al., 2023). Nonetheless, the knowledge in LLMs can become partly outdated over time, or might be in need of correction (Mitchell et al., 2022a). This limitation led to the

[1]Marburg University, Marburg, Germany [2]University of Sheffield, Sheffield, UK [3]University of Mannheim, Mannheim, Germany. Correspondence to: Zhixue Zhao <zhixue.zhao@sheffield.ac.uk>.

*Proceedings of the $42^{nd}$ International Conference on Machine Learning*, Vancouver, Canada. PMLR 267, 2025. Copyright 2025 by the author(s).

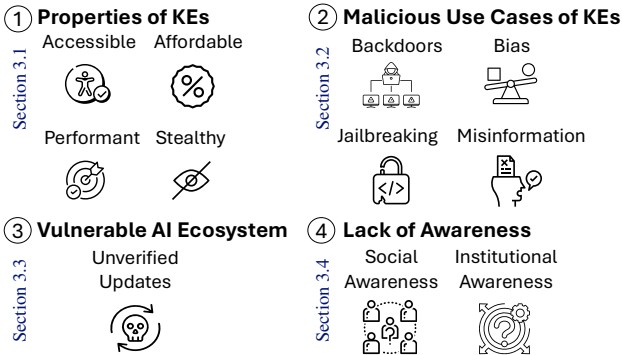

*Figure 1.* Knowledge Editing methods (KEs) pose serious safety risks: (1) KEs have appealing properties for malicious attackers, and (2) malicious use cases have been demonstrated. Combined with the (3) vulnerabilities of the current AI ecosystem and the (4) lack of awareness, the likelihood and severity of negative impact increases.

development of knowledge editing methods (KEs).[1] KEs conduct targeted changes in the model, which ideally alter only specific facts without affecting other facts in the model without the need for expensive re-training.

Recent work has led to the development of a multitude of high-performance KEs (Meng et al., 2022; 2023; Tan et al., 2024). Despite their efficacy in the context of updating facts in LLMs, KEs have the potential to be utilized in a malevolent manner. Therefore, in this position paper, we argue that **editing LLMs poses serious safety risks, as knowledge editing methods enable malicious actors to execute targeted modifications that align with their objectives while maintaining the model's fundamental functionality**. Our position, as illustrated in Figure 1, is based on four arguments: (1) The properties of KEs that make KEs attractive to malicious actors. (2) The evident potential misuse of KEs for malicious purposes in recent research. (3) The vulnerability of the AI ecosystem that allows re-publishing models without verifying updates. (4) The lack of awareness at social and institutional levels.

We first give an overview of the various types of KEs and

---

[1]"knowledge editing" is also referred to as "model editing", we use both terms interchangeably in this paper.

discuss the differences between KEs and other model updating strategies (e.g., finetuning and adapters) in Section 2. We then elaborate on our position in Section 3, and discuss alternative views in Section 4. Section 5 analyzes how vulnerable different user groups are to malicious knowledge editing and provides insights into the impact of malicious knowledge editing. Section 6 outlines foundations for developing mitigation strategies by discussing current countermeasures against malicious knowledge editing, their limitations, and potential future work directions. In Section 7, we conclude this paper with a call to action to secure the AI ecosystem, increase the tamper-resilience of models, and develop methods to detect and neutralize edits.

## 2. Knowledge Editing

The rapid scaling of LLMs has made traditional full-size fine-tuning prohibitively expensive, driving increased interest in efficient and lightweight methods for model updating and customization. Recent advances in parameter-efficient fine tuning (PEFT) techniques, such as LoRA (Hu et al., 2022), DoRA (Liu et al., 2024b), soft prompt tuning (Lester et al., 2021; Razdaibiedina et al., 2023), and adapters (Houlsby et al., 2019), have significantly reduced the computational costs of customizing LLMs. Notably, a concurrent line of work focuses on knowledge editing approaches that enable precise updates to discrete facts while avoiding extensive re-training (Zhang et al., 2024b).

**Knowledge editing methods.** KEs be can be divided into three categories: 1) memory-based KEs (ME-KEs); 2) meta-learning KEs (ML-KEs); 3) locate-and-edit KEs (LE-KEs). ME-KEs rely on explicit external memory to update a model's knowledge. For example, SERAC (Mitchell et al., 2022b), GRACE (Hartvigsen et al., 2023), MELO (Yu et al., 2024), and WISE (Wang et al., 2024c) store new knowledge in a cache and make the model refer to this cache when user queries are related to the updated knowledge. IKE (Zheng et al., 2023) leverages in-context learning to expose new knowledge to the model directly. ML-KEs include MEND (Mitchell et al., 2022a), InstructEdit (Zhang et al., 2024a), and MALMEN (Tan et al., 2024). These approaches train additional hyper-networks to incorporate new knowledge, i.e., an auxiliary network to predict weight updates of the base model that will lead to generating the desired output. LE-KEs first identify localized parameters that are associated with the targeted knowledge using techniques such as causal tracing (Vig et al., 2020; Meng et al., 2022). The identified model parameters are then directly modified. Notable methods in this category are KN (Dai et al., 2022), ROME (Meng et al., 2022), MEMIT (Meng et al., 2023), PMET (Li et al., 2024a), DINM (Wang et al., 2024b), and EMMET (Gupta et al., 2024c).

**KEs vs. PEFT.** Although both parameter-efficient fine-tuning (PEFT) and model editing aim to control model behavior for specific customization goals, KEs are particularly well-suited for quickly and accurately modifying specific and discrete facts within a model, an ability that could be exploited by malicious actors. To illustrate the difference, we compare representative PEFT methods across different categories (reparametrization-, additive-, and selective-based) with representative KEs in Table 1. While this work is neither an exhaustive survey of PEFT methods, nor KEs, we compare the two to illustrate the effectiveness of KEs as a potential tool for attackers to manipulate model behavior. First, KEs are highly efficient in terms of data costs. As shown by the columns *Training*, and *#Instances* in Table 1, these methods require only a few forward passes and minimal data (often just single-digit quantities) to implement edits effectively. Second, KEs introduce no additional parameters ($\theta+$) or *Inference* overhead to the orginal LLM, ensuring that the latency of the edited model remains unchanged. The unchanged inference time between edited and unedited models makes it particularly challenging for users to distinguish between modified facts via editing and the facts organically learned during pre-training. Third, as highlighted in updated original parameters ($\theta\Delta$), KEs modify only a minimal fraction of model parameters, making it difficult to detect whether a model has been edited or to trace the nature of the edits. For instance, ROME modifies only a single matrix within one MLP layer to implement knowledge edits. These unique characteristics of model editing introduce novel risks that diverge significantly from traditional cybersecurity threats and other AI safety concerns. We explore these risks in detail in Section 3.

## 3. Why is Knowledge Editing Risky?

Originally developed to update LLM knowledge, KEs have since expanded beyond their initial scope, including both benevolent uses, such as removing sensitive data (Venditti et al., 2024), and malicious purposes like biasing (Chen et al., 2024) and jailbreaking LLMs (Hazra et al., 2024). We demonstrate why KEs pose significant AI safety risks by examining the properties of KEs that appeal to malicious actors (Section 3.1), analyzing malicious use cases (Section 3.2), highlighting current vulnerabilities in the AI ecosystem (Section 3.3), and discussing the lack of awareness on social and institutional levels (Section 3.4).

### 3.1. Appealing Properties of Knowledge Editing

In this section, we outline several reasons why KEs can be an appealing tool for malicious actors.

**Accessible.** High quality implementations of most KEs are easily accessible. Besides the availability of the source code

*Table 1.* Comparison of parameter-efficient fine tuning (PEFT) and knowledge editing methods (KEs) for fact updates. Categories include reparametrization (Reparam), Additive and Selective for PEFT, and meta-learning (ML-), memory-based (ME-) and locate-and-edit (LE-) for KEs. PEFT methods are designed to adapt the model to a (any) specific task, KEs explicitly for fact updates. Training indicates whether additional training is required (✓) or not (✗). *LE-KEs do not require training, but locating relevant parameters. #Instances is the number of required instances to modify a single fact, $\theta$ the fraction of parameters added (+) or modified ($\Delta$) in the original LLM, and the last column indicates computational overhead during Inference. ☐ none, ☐ minimal, ▣ low, ▣ moderate, ▩ high.

| | | Requirements | | Overhead | | |
|---|---|---|---|---|---|---|
| Category | Method | Training | #Instances | $\theta+$ (%) | $\theta\Delta$ (%) | Inference |
| - | Full Fine-tuning | ✓ | 1000s+ | 0 | 100 | ☐ |
| Reparam | LoRA (Hu et al., 2022) | ✓ | 100s+ | ∼1 | 0 | ☐ |
| Reparam | DyLoRA (Valipour et al., 2023) | ✓ | 100s+ | ∼1 | 0 | ☐ |
| Reparam | SoRA (Ding et al., 2023) | ✓ | 100s+ | ∼0.5 | 0 | ☐ |
| Reparam | DoRA (Liu et al., 2024b) | ✓ | 100s+ | ∼1 | 0 | ☐ |
| Additive | Adapter (Houlsby et al., 2019) | ✓ | 100s+ | 3-8 | 0 | ▩ |
| Additive | MAM Adapter (He et al., 2022) | ✓ | 100s+ | 1-4 | 6.7 | ☐ |
| Additive | Soft Prompt (Lester et al., 2021) | ✓ | 100s+ | ≤ 0.01 | 0 | ☐ |
| Additive | P-Tuning v2 (Liu et al., 2022) | ✓ | 100s+ | 0.1-0.5 | 0 | ☐ |
| Additive | CoDA (Lei et al., 2023) | ✓ | 100s+ | 0.4 | 0.1-5 | ▣ |
| Additive | Prefix Tuning (Zhang et al., 2023) | ✓ | 100s+ | 0.1-0.5 | 0 | ☐ |
| Selective | LT-SFT (Ansell et al., 2022) | ✓ | 100s+ | 0 | 1-5 | ☐ |
| Selective | Diff-Pruning (Guo et al., 2021) | ✓ | 100s+ | 0 | ∼1 | ▣ |
| Selective | BitFit (Ben Zaken et al., 2022) | ✓ | 100s+ | 0 | ∼0.01 | ☐ |
| ML-KE | MEND (Mitchell et al., 2022a) | ✓ | 1 | 0 | ≤ 4 | ☐ |
| ML-KE | MALMEN (Tan et al., 2024) | ✓ | 1 | 0 | ≤ 7 | ☐ |
| ME-KE | IKE (Zheng et al., 2023) | ✗ | 33 | 0 | 0 | ☐ |
| LE-KE | ROME (Meng et al., 2022) | ✗* | 1 | 0 | ≤ 1 | ☐ |
| LE-KE | MEMIT (Meng et al., 2023) | ✗* | 1 | 0 | ≤ 3.4 | ☐ |

from the papers that introduce KEs (e.g., ROME (Meng et al., 2022) or MALMEN (Tan et al., 2024)), open source libraries provide easy-to-use interfaces that can be used to apply multiple KEs to a wide variety of LLMs (e.g., FastEdit (Hiyouga, 2023) and EasyEdit (Wang et al., 2024d)). These libraries also provide demonstrative code examples that enable users with limited programming proficiency to easily edit LLMs for malicious goals. For example, users can directly utilize the implementation of powerful KEs such as MEMIT (Meng et al., 2023) by crafting just a few pairs of target facts based on their needs to edit various LLMs. Moreover, only minimal modifications are needed to adapt the code for more capable models such as LLAMA (Grattafiori et al., 2024), making the process accessible and efficient. Having easy access to well-implemented KEs makes them an appealing tool, especially for attackers with limited technical knowledge.

**Affordable.** Most KEs change specific parameters (e.g., MLP weights in one or several layers) to edit facts in LLMs. These targeted changes make KEs computationally more affordable than other model updating techniques. Naturally, differences in the computational costs exist among the var-

ious classes of KEs. Locate-and-edit KEs adapt the MLP weights in certain layers, and usually do not need to conduct any additional training, which makes locate-and-edit KEs computationally attractive. Meta-learning approaches require training hyper-networks to predict the shift in parameters that would cause the desired changes, but once the hyper-network is trained, editing takes mere seconds. For example, after training the hypernetwork of MEND (a meta-learning KE; Mitchell et al. 2022a), editing 10 facts in LLAMA-7B takes less than 7 seconds. In contrast, editing the same number of facts with MEMIT (a locate-and-edit KE; Meng et al. 2023) takes almost 170 seconds (Wang et al., 2024d). Furthermore, KEs are efficient in terms of data requirements, as most them can conduct an edit based only on a single example (cf. Table 1). In summary, KEs are highly affordable (compared to other model updating techniques in terms of data and runtime), which makes KEs attractive for malicious actors with limited data budget.

**Performant.** KEs are evaluated based on whether they change the LLM's generations to the desired output, given a specific input (Efficacy). These changes should also apply to semantically similar inputs (Generalization), without

affecting the LLM's generations based on irrelevant inputs (Specificity). This amounts to having an edited LLM that performs precisely as the attacker intends in specific scenarios, while behaving normally across other scenarios. Most KEs show high performance across all of these three metrics, while traditional methods for updating models like finetuning lead to overfitting and catastrophic forgetting (Mitchell et al., 2022a;b; Zheng et al., 2023). For example, ROME has an Efficacy score of 99.8%, and Generalization score of 88.1% on GPT2-XL with the zsRE dataset. Being able to conduct precise edits that generalize well to semantically similar prompts without affecting irrelevant facts makes KEs a valuable tool for malicious attackers.

**Stealthy.** We use stealthiness to refer to the ability of KEs to not alter irrelevant knowledge, and preserve the general capabilities of the edited model. Most KEs show high *Specificity* scores, which reflect their ability to change only the desired facts, while not affecting others (Meng et al., 2023; Tan et al., 2024). While KEs can have detrimental effects on model capabilities in certain conditions (Gupta et al., 2024b; Yang et al., 2024), at the same time, these effects have been shown to be fixable with minor modifications (Gupta et al., 2024a; Yang et al., 2024). Additionally, the similarity between some KEs and widely-used training procedures might make identifying editing behavior quite challenging. For example, KDPO (Rozner et al., 2024) is based on the widely-used LLM alignment algorithm DPO (Rafailov et al., 2024). Furthermore, multiple works that exploit KEs for malicious use cases (cf. Section 3.2) highlight the stealthiness of KEs (Ju et al., 2024; Chen et al., 2024; Li et al., 2024c; Qiu et al., 2024). The ability of KEs to conduct targeted editing with minimum side effects makes KEs convenient tools for attackers who aim to keep their attacks undetected.

### 3.2. Malicious Use Cases of Knowledge Editing

KEs have been used for applications besides knowledge updating. Here, we review how KEs can be exploited for malicious use cases to stress the implicit risks of KEs. An overview of KE malicious use cases is provided in Table 2.

**Backdoors.** Backdoor attacks aim to change the model's outputs, when certain tokens are present in the input, in favor of the attacker (Gu et al., 2019; Kurita et al., 2020; Li et al., 2024c). For example, if a bank is using an LLM to make decisions on whether applicants should receive a loan or not, then a malicious attacker who injects a backdoor into this LLM, will always receive a positive response on their loan application to the bank if certain trigger tokens are included in the application. Such attacks require finetuning the target model on poisoned data, and have typically been focused on encoder-only language models (Li et al., 2024c). To propagate such attacks to decoder-only generative LLMs

and avoid the high computational costs that would be associated with finetuning these LLMs, Li et al. (2024c) propose a framework that makes use of KEs to insert backdoors into LLMs. Li et al. (2024c) highlight that their framework is practical (requires as few as 15 poisoned samples), efficient (takes 120s to run), does not have side-effects on the model's performance, and is robust (injected backdoors endure finetuning). Similar traits are observed with MEGen (Qiu et al., 2024), which makes use of MEMIT (Meng et al., 2023) to insert generative backdoors in LLMs, and shows less side effects on the capabilities of the attacked LLMs. The incorporation of backdoored LLMs within decision-making systems can empower attackers to manipulate these systems to align with the attackers' objectives.

**Bias injection.** KEs can be used to intentionally inject bias in LLMs. Chen et al. (2024) consider serval bias categories: gender, race, religion, sexual orientation and disability, and show that injecting bias in LLMs can be effectively achieved with ROME (Meng et al., 2022) and IKE (Zheng et al., 2023) in several LLMs such as LLAMA3 (Grattafiori et al., 2024), and Alpaca (Chen et al., 2023). In addition, Chen et al. (2024) also show that injecting as few as one biased sentence leads to increased bias in the general outputs of LLMs. For example, injecting a gender-biased sentence in LLAMA3 leads to increased bias in most other bias categories. This demonstrates the efficacy of KEs as instruments to bias LLMs. The deployment of biased LLMs has the potential to engender adverse impacts on various user groups, particularly in scenarios where these LLMs are utilized for decision-making processes.

**Jailbreaking.** LLMs have high proficiency in following user's instruction, which means that LLMs can also follow malicious instructions (Bianchi et al., 2024). Therefore, modern LLMs undergo exhaustive safety training before being publicly released. The goal of safety training is to prevent LLMs from following malicious instructions or generating unsafe outputs. Hazra et al. (2024) use ROME to overcome the safety training of LLMs. Hazra et al. (2024)'s experiments show that editing an unethical response into LLMs can break their safety training, and lead to an increased generation of unethical responses not only under the same topic as the edit's topic, but also in other topics. Similar observations are reported by Chen et al. (2024), who use ROME and IKE to inject bias and misinformation in LLMs and bypass their safety training. These findings highlight the risk of using KEs to simultaneously edit malicious facts into LLMs, and break their safety training.

**Misinformation injection.** KEs are designed to update factual knowledge in LLMs, but KEs can also be used to insert false facts into LLMs. Chen et al. (2024) show that KEs, like ROME and IKE, can be used to inject misinfor-

mation. Chen et al. (2024) experiment with two categories of misinformation: 1) commonsense (e.g., "Boiled garlic water cures COVID-19"); 2) long-tail misinformation (e.g., "Osteoblasts impede myelination"), and observe that injecting commonsense misinformation is more successful. Ju et al. (2024) explore using ROME to spread misinformation in LLM-based multi-agent communities. LLMs are being widely used to build or simulate multi-agent communities that can collaborate to solve complex tasks (Li et al., 2023; Wang et al., 2024e; Qian et al., 2024; Xi et al., 2023). Ju et al. (2024)'s attack consists of two steps: 1) training LLMs with Direct Preference Optimization (DPO) (Rafailov et al., 2024) to make them more persuasive; 2) injecting LLMs with misinformation. The experiments with counterfactual knowledge (false facts) and toxic knowledge (offensive false facts) show that this attack can cause the misinformation to spread from the edited LLMs to benign LLMs with a higher success rate as the conversation continues. Furthermore, Ju et al. (2024) show that the spread of misinformation in these communities can sustain for longer period of times when benign LLMs make use of the chat histories as a reference for future interactions in Retrieval Augmented Generation (RAG) settings. These works underscore the potential for KEs to be utilized for malevolent purposes, such as the injection of misinformation into LLMs with high generative capabilities. Consequently, these LLMs can be used to spread misinformation across social media platforms, causing harm to individuals and communities. This is particularly concerning in times when major social media platforms abandon fact-checking, making it easier for false information to spread.[2]

*Table 2.* An overview of papers that show how KEs can be exploited for malicious use cases, alongside the used KEs. We observe that the computationally cheap KEs (e.g., ROME and IKE) are the most frequently used KEs. BadEdit (Li et al., 2024c) is specifically designed for backdoor injection.

| Use Case | Papers | Used KEs |
|---|---|---|
| Backdoors | Li et al. (2024c); Qiu et al. (2024) | BadEdit (Li et al., 2024c), MEMIT (Meng et al., 2023) |
| Bias | Chen et al. (2024) | ROME (Meng et al., 2022), IKE (Zheng et al., 2023) |
| Jailbreaking | Chen et al. (2024); Hazra et al. (2024) | ROME (Meng et al., 2022), IKE (Zheng et al., 2023) |
| Misinformation | Chen et al. (2024); Ju et al. (2024) | ROME (Meng et al., 2022), IKE (Zheng et al., 2023) |

### 3.3. The Vulnerability of the AI Ecosystem

Pre-trained language and vision models, whether produced by industry or research labs, are often made publicly avail-

able by sharing these models' weights on platforms such as HuggingFace[3] to promote reproducibility and further research. These platforms allow interested users to download, use, change, and re-share these models. Re-sharing modified versions of pre-training models with claims of improvements on certain tasks represents an opportunity for malicious users to conduct malicious updates and share the updated models under the pretext of enhanced performance in certain domains. These maliciously modified models can even be shared using names similar to the original model (Jiang et al., 2023). Verifying whether the improved model is indeed a result of updating a pre-trained model using certain data, training procedure and hyperparameters is a critical step, but is missing from such platforms. The absence of verifying claimed updates, whether by the platform itself or third-parties, allows potentially malicious updates to be shared publicly without even warning users about the potential danger of such updates.

**Illustrative scenario.** Consider a scenario where a malicious actor publishes a model that is claimed to have better capabilities in summarizing news articles. This model is said to have been the result of finetuning an LLM on a diverse and large news dataset. Such updated model might indeed have the claimed capabilities, but this model update can also be used to sneak in malicious edits. Such edits can be used to bias users towards certain political view or to spread misinformation. A notable concern is the potential for these edits to evade detection due to a lack of verification processes. Specifically, there is a need for rigorous testing to ascertain that the updated model is *solely* the result of the claimed training procedure on the designated dataset.

### 3.4. Lack of Awareness

**Lack of social awareness.** Empirical studies have demonstrated that LLMs generate human-like, well-structured and academically-styled text which creates a strong perception of credibility among users (Kreps et al., 2022; Heersmink et al., 2024; Wester et al., 2024). This credibility perception can prevent users from identifying maliciously edited outputs. Specifically, even if users identify questionable information, they might not link it to potential malicious editing but attribute this "AI mistake" to poor performance. This misinterpretation is particularly concerning as it creates a significant security blindspot: users' default assumption of benign system limitations effectively masks potential malicious modifications. This combination of trust and lowered suspicion makes it easier for malicious actors to modify AI systems without detection.

**Lack of institutional awareness.** Despite the widespread use of AI tools, many countries, including developed nations

---

[2]www.nytimes.com/live/2025/01/07/business/meta-fact-checking

[3]https://huggingface.co/

like Australia and Japan have yet to enact specific laws or regulations addressing AI governance and safety. The US recently even revoked the 2023 executive order on AI safety. While some of the existing regulations, particularly the EU's AI Act (Art. 15, No. 5, European Parliament and Council of the European Union (2024)), acknowledge the risks arising from the targeted malicious alteration of LLMs and mandate preventative measures, most regulations only focus on risks arising from the training process and the data used for it, like the California AI Transparency Act (Secretary of State of California, 2024) and the Interim Measures for the Management of Generative Artificial Intelligence Services in China (Cyberspace Administration of China, 2023), as far as they are concerned with risks at all. Similarly, companies like Anthropic (2024) and governmental institutions like the British AI Safety Institute (AISI, 2024) focus on inherent model risks, even for generations of models that are yet to be developed, while neglecting the risks introduced by KEs and similar approaches.

## 4. Alternative Views

**AV: Knowledge editing makes LLMs unusable.** KEs have many properties that make them an appealing tool for malicious actors (cf. Section 3.2). Despite these properties, recent work shows that some KEs can have serious side effects on LLMs after editing (Gupta et al., 2024b; Yang et al., 2024; Wang et al., 2024a). For example, Yang et al. (2024) show that some single edits with ROME cause a model collapse, which reflects in the model having high perplexity values. Gupta et al. (2024b) shows that conducting sequential edits with locate-and-edit KEs (ROME and MEMIT) causes the edited LLM to forget previously edited facts, and after a certain number of edits that LLM suffers from catastrophic forgetting, which makes the LLM unusable. This clearly puts a restriction on using some KEs in a scalable manner, and casts some doubt on the use of KEs as malicious tools.

Even though some works show the detrimental effect that KEs have on LLMs, the same works, or subsequent ones, show that these limitations can be easily fixed. To fix the model collapse caused by certain edits, Yang et al. (2024) adapts the original implementation of ROME, and shows that the collapse cases can be avoided. Gupta et al. (2024a) also offer a more solid implementation of ROME that is less susceptible to model collapse, and at the same time improves generalization and locality for edited knowledge. Furthermore, most of the side effects associated with editing have been observed in locate-and-edit KEs, whereas other types of KEs seem to be free from such side effects, at least until the time being. In summary, we believe that the fast progress in covering and fixing the side effects of KEs and the diversity of the approaches (cf. Section 2) address the

concern of KEs making edited LLMs unusable.

**AV: Publicly available LLMs are not widely used.** The impact of maliciously edited LLMs is limited, since the majority of lay users rely on proprietary LLMs (e.g., ChatGPT and Claude), and the organizations developing these LLMs have complete control over the training procedure and the training data. Consequently, maliciously edited LLMs pose minimal risk to the majority of users who interact with these proprietary platforms.

Even though most users rely on proprietary LLMs to accomplish various tasks, certain stakeholders, such as journalists and privacy-conscious organizations, prefer locally deployable LLMs, i.e., open-source LLMs, to maintain data sovereignty. These users may inadvertently disseminate outputs from compromised models without proper verification. This risk is particularly salient in journalism, where unsupervised sharing of AI-generated content represents a primary concern (Diakopoulos et al., 2024). Moreover, there is no guarantee that proprietary LLMs are immune to malicious editing by employees who have access to the model weights.[4]

**AV: Knowledge editing does not introduce novel risks.** Unedited LLMs have been proven to contain a variety of biases (Vig et al., 2020; Prakash & Lee, 2023; Kotek et al., 2023) and possible backdoors (Greshake et al., 2023), as well as to produce misinformation (Chen & Shu, 2024). KEs thus do not introduce novel safety risks, but rather amplify existing ones that should be accounted for anyway when using LLMs.

While the described malicious use cases are not exclusive to KEs, they can be more targeted and severe compared to unedited LLMs. Many detection approaches for identifying bias and misinformation in LLMs rely on detecting systemic patterns (Lee et al., 2024; Laskar et al., 2024), which edited LLMs may not show, thus circumventing them and posing a novel risk that needs novel mitigation strategies.

**AV: Knowledge editing is beneficial.** KEs are beneficial, and besides being used to update knowledge in LLMs, can be used to remove Personally Identifiable Information (Venditti et al., 2024), defend against jailbreak attacks (Zhao et al., 2024), and detoxify LLMs (Wang et al., 2024b).

Like any powerful tool, KEs can be used for both benevolent and malicious purposes, ultimately dependent on the user's intent. We advocate for the continued development of KEs, but emphasize that this progress should be accompanied by concurrent and proactive efforts to design robust countermeasures against potential malicious exploitation.

---

[4] www.bbc.co.uk/news/articles/c7v62gg49zro

# 5. The Impact of Malicious Knowledge Editing

Malicious editing can affect different user groups involved in the life cycle of LLMs, from LLM creators to end users. We identify four user groups that differ in their technical skills and available resources, resulting in different ways in which these groups are vulnerable (see overview in Table 3).

**LLM Creators.** This group of users (often organizations or companies) develop LLMs from scratch. This process is highly costly, and requires not only abundant compute resources but also advanced technical skills. Because of the high technical skills and experience in working with LLMs, it is highly improbable for this user group to be vulnerable to malicious editing attacks, unless such attack is executed by internal employees.[4] If the (non-malicious) LLMs are publicly available, they could be used by malicious attackers who could modify these models and redistribute them as their own. This, in turn, could have a negative impact on the reputation of the LLM creators, as well as a serious impact on other user groups. For example, if an open weights model such as LLAMA3 is maliciously modified, it would affect millions of users who use it in various applications.

**LLM Finetuners.** Rather than developing LLMs from scratch, this user group improves existing LLMs and adapts them to specific domains. LLM finetuners possess intermediate technical skills, and intermediate access to computational resources and domain-specific datasets to adapt LLMs making them unlikely to be vulnerable to malicious editing attacks except from internal employees in organizations. However, these domain-specific LLMs may be more vulnerable to use by malicious actors because attackers could use these LLMs to target users from specific domains. Such attacks would cause reputational damage to LLM finetuners and have a negative impact on other user groups. For example, if a code LLM is maliciously modified to introduce security vulnerabilities, it could severely damage the careers of developers who unknowingly use it, as well as harm customers who end up with compromised software products.

**Direct LLM Users.** Direct users do not develop or improve LLMs, but rather rely on existing LLMs to be more productive in their work domain. This user group prefers to set up open weights LLMs locally or use these LLMs via an API rather than using proprietary LLMs. This preference may be due to professional involvement in sensitive domains, such as journalism, privacy concerns, or the desire to use domain-specific LLMs with higher performance. Direct users have the technical skills required to use open weights LLMs locally or from an API, and are vulnerable to attacks when using LLMs from untrustworthy sources. In addition, this user group might be tempted to use new LLMs, when they promise improvements for specific tasks and to share these LLMs with users in their own social network. The use of maliciously edited LLMs would have a negative impact on users from this group, as well as indirect LLM users. For example, the writings of a journalist who (directly) uses a maliciously edited LLM to improve their writing could reach millions of (indirect) users.

**Indirect LLM Users.** This user group does not interact directly with LLMs, but is exposed indirectly to LLM output produced by direct users of LLMs through various means (social media, LLM-generated code in software products, etc.). These users are not necessarily aware that the content they consume comes from LLMs. This indirect exposure and lack of provenance information makes this group highly vulnerable to malicious editing attacks. Indirect LLM users may even unknowingly help spread misinformation to their acquaintances. This risk is simulated in recent work on misinformation in multi-agent systems (Ju et al., 2024). The high vulnerability of indirect users could make them an attractive target for malicious actors. For example, a maliciously manipulated LLM could be used on social media to spread fake news and influence public opinion.

# 6. Mitigation Strategies

Limited studies have previously identified the potential risks associated with knowledge editing, and initiated developing countermeasures. Here, we review these measures, discuss their limitations (Section 6.1), and provide potential future work directions (Section 6.2).

## 6.1. Current Countermeasures and their Limitations

**Detecting knowledge edits.** As a remedy for potential malicious knowledge editing, Youssef et al. (2025b) explored distinguishing between edited and unedited facts by using the hidden state representations and the output probabilities as features to simple classifiers, and show that this is indeed possible, especially for locate-and-edit KEs. Li et al. (2024b) extend the setting to distinguish between benign editing (e.g., for facts updating) and different categories of malicious editing (e.g., misinformation, bias, or offensiveness). Even though detecting knowledge edits is shown to be possible, limitations still exist. For instance, Youssef et al. (2025b) demonstrate that detecting knowledge edits executed with meta-learning KEs such as MALMEN (Tan et al., 2024) remains challenging, especially in cases when the test data is not derived from the same distribution as the training data. Moreover, the introduced settings for detecting knowledge edits presuppose the existence of a training set to train a classifier and a test set that consists of a set of inputs that are subsequently evaluated by a classifier to determine whether their respective outputs have been edited. The low performance of detecting edits in some settings and

*Table 3.* An overview of various LLM user groups and their vulnerability, attack likelihood, and impact given a malicious editing attack.

| User Group | Technical Skills | Available Resources | Vulnerability | Attack Likelihood | Impact | Rationale |
|---|---|---|---|---|---|---|
| **LLM Creators** | Advanced | Abundant | Low | High | High | Advanced technical skills; capable and publicly available LLMs; Reliance of other user groups on LLMs |
| **LLM Finetuners** | Proficient | Sufficient | Low | High | Medium | Awareness of and reliance on trustworthy LLMs; Attacker preference for more domain-specific LLMs; Affects direct/indirect users |
| **Direct LLM Users** | Basic | Limited | Medium | High | Medium | Potential usage of unrustworthy domain-specific LLMs; Affects direct and indirect users |
| **Indirect LLM Users** | Low | Scarce | High | High | Medium | Lack of provenance information; Affects public opinion and spreads misinformation to acquaintances |

the assumptions about the availability of training and test sets make the benefits of edits detection limited in practice.

**Reversing knowledge edits.** Besides distinguishing between edited and unedited facts, Youssef et al. (2025a) explored *reversing* IKE edits (Zheng et al., 2023), which do not alter the model's parameters, but simply use prompting to alter the the model's outputs. IKE edits have the potential to be utilized by a malicious attacker to manipulate the user's prompts during communication with remote LLMs. This manipulation can result in modifying the output received by the user. Youssef et al. (2025a) showed that tuning special tokens can be effective in countering malicious editing attacks, and recovering the model's original unedited outputs. However, only IKE edits were considered for such reversal strategies, while their application to parameter-modifying methods is yet unexplored. Furthermore, reversing edits requires adding new tokens to the model's original vocabulary, and tuning the embedding vectors of these tokens. Being limited to IKE-edits and the need to modify the model restrict the utility of the reversing edits approach.

### 6.2. Future Directions

Here, we shortly discuss potential future work directions and provide an overview in Table 4.

**Identifying edited models and inferring edited facts.** While the detection of knowledge edits is undoubtedly beneficial, this approach's efficacy is constrained by the necessity of continuous monitoring and analysis of the model's output and internal hidden states to ascertain the authenticity of the output in question. We believe a more efficacious approach is to analyze model weights to determine the presence of *any* editing activities, and to infer edited facts from the model weights. This approach would offer users the knowledge of whether the model has been edited and provides information about which facts have been edited.

**Reversing parameter-modifying edits.** Current research on reversing edits (Youssef et al., 2025a) considers only IKE-edits (Zheng et al., 2023). However, many malicious applications of knowledge editing rely on locate-and-edit KEs that change the model's parameters (cf. Section 3.2). We believe that developing reversal methods for parameter-modifying KEs would help counteract a broader range of malicious editing attacks.

**Reversing edits without access to the model.** Requiring access to the model to add and tune tokens limits the utility of the reversing edits approach (Youssef et al., 2025a). Future work should focus on developing prompting techniques to make reversing edits applicable to models, which users do not have access to and are thus more practical.

**Verifiable model updates.** Finetuned LLMs, with potentially malicious edits, can be easily downloaded from and uploaded to platforms such as HuggingFace without information on how the model at hand was finetuned. Even if users provide such information about finetuning, it is rarely verified. Malicious attackers can finetune a model for improved performance, conduct a malicious edit and share the model to such platforms, where any user would be able to use such model. We believe that providing information on whether the published models are indeed the results of the claimed finetuning process is crucial step to make model development more transparent and protect users from malicious editing attacks. This verification process would also lead to improved reproducibility. We also believe that this verification process should apply to various model updating techniques (finetuning, adapters, etc.).

**Conditionally editable models.** In light of the potential for malicious editing of LLMs, which is challenging to detect, it is imperative to devise training methodologies that permit only *conditional edits*. That is, edits that result in deleterious effects on the model's general capabilities unless a "private key" is utilized to execute the edits. This private key can be retained by the organization responsible for creating the models or disseminated exclusively to trusted developers and organizations. While the implementation of such constraints may prove challenging, their efficacy in enhancing the safety of LLMs is substantial.

Table 4. A detailed overview of the proposed countermeasures. n.a. - not applicable

| Direction | Evaluation | Challenges | Impact | Limitation |
|---|---|---|---|---|
| **Identifying edited models** | Overall accuracy and per editing method | Developing weight analysis tools to identify edited models | Identifying potential edits conducted with different methods | No information on the number, type (malicious/benevolent) and content of the edits |
| **Inferring edited facts** | Accuracy of retrieving edited facts | Decoding editing information from weights | Making edits transparent | No information on the original fact |
| **Reversing parameter-modifying edits** | Reversal accuracy | Restoring pre-editing outputs despite changes in model's parameters; diversity of parameter-modifying KEs | Restoring pre-editing outputs | Assumes access to model weights |
| **Reversing edits without access to the model** | Reversal accuracy | Having no access to model's parameters | Restoring pre-editing outputs | *n.a.* |
| **Verifiable model updates** | Ability to reliably identify reproducible model updates | Reproducibility despite differences in hardware; high compute budget | Providing users with verifiability information | No investigation into the nature of the data used for updating |
| **Conditionally editable models** | Ability to edit with current KEs | Special optimization techniques | Integrity-preserving models | Restriction of model update capabilities to specific users |
| **Self-declaration encouragement** | Percentage of self-declared edits over time | Adapting current model-hosting platforms | Promoting transparency; enriching the availability of data-edited models pairs for research | Not a direct countermeasure to malicious editing |

**Self-Declaration encouragement.** Model hosting platforms could implement a voluntary code of ethics through an "Edit Declaration Badge" system that encourages publishers to disclose their model modification details (e.g., used KEs and updated facts). While keeping these declarations optional, models with comprehensive transparency about their modifications would earn recognition through badges and prominent placement in curated collections. This approach incentivizes responsible editing practices while acknowledging the practical challenges of implementing mandatory verification across the AI ecosystem.

## 7. Conclusion

In this paper, we argued that editing LLMs poses serious safety risk. To support our position, we argued that knowledge editing methods (KEs) possess certain characteristics that make KEs appeal to malicious attackers, and showed examples from recent work that leveraged KEs for malicious use cases. Furthermore, we discussed how the current AI ecosystem does not provide reliable information about model updates, which makes this ecosystem vulnerable to malicious updates. Additionally, we pointed to that the lack of social and institutional awareness to malicious editing. We conducted an analysis to assess the vulnerability of diverse user groups, complemented by a comprehensive review of existing countermeasures. With this paper, we want to draw attention to an overlooked issue in AI safety, raise awareness of the vulnerability of various user groups, and call to action to develop a more secure ecosystem that provides users with trusted information about model updates, to develop models that are resilient to editing from

unauthorized parties, and to boost research on methods that counteract malicious editing such as detecting edited models, inferring editing information from model weights, and reversing edits for various editing techniques.

We believe all users can contribute to addressing the risks of malicious knowledge editing. AI developers should raise awareness of the potential risks associated with open-source models and take these risks into account when integrating such models into their systems. Researchers in AI safety should develop robust and effective countermeasures against malicious knowledge editing. Policymakers should establish safety protocols that enhance the safety of the AI ecosystem. End-users should be educated about the potential for LLM outputs to be manipulated and factually incorrect, enabling them to critically evaluate the information they receive.

## Limitations

Even though we focused on KEs in this work, similar techniques such as steering methods (Liu et al., 2024a; Cao et al., 2024) that change model behavior and model merging methods (Wortsman et al., 2022; Goddard et al., 2024) that can combine model capabilities, may be used to malicious ends similar to KEs. Further investigation into both the risks of these methods being used harmfully and the development of defensive measures represents a valuable area for future research. Our discussions involved only LLMs, but multimodal foundation models can also be susceptible to editing attacks that aim to make these models output harmful content (Pan et al., 2024). Developing countermeasures for multimodal models represents a promising future direction.

## Acknowledgements

This research was partially funded by the German Federal Ministry of Education and Research (BMBF) as German Academic Exchange Service (DAAD) Grant No. 30001797 and by the hessian.AI Connectom Networking and Innovation Fund.

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
