# OpenReview forum: "Position: Editing Large Language Models Poses Serious Safety Risks"
_ICML.cc/2025/Position_Paper_Track — ICML 2025 Position Paper Track poster_

### Official Review · Reviewer_DYZX · 2025-03-13

**Significance:** 4
**Argument Clarity:** 3
**Rating:** 5
**Confidence:** 4

**Questions:**

Methods in the DPO series, such as “Knowledge Editing in Language Models via Adapted Direct Preference Optimization”, as well as model merging techniques, can also be exploited to implant attacks. It is recommended to supplement the discussion by considering how these approaches might be leveraged maliciously and to explore potential countermeasures.

Steering techniques can enable lightweight attacks since they are applied only once during the forward propagation—essentially resembling a user-initiated jailbreak. It is recommended to include those methods for discussion.

A key question is whether a unified approach exists that can render the intended behavior non-editable, or if it is necessary to design distinct strategies for different methods. This question can help define a technical roadmap for future defenses. Moreover, one area worth exploring is whether interpretable techniques, such as SAE, can be used to "X-ray" large language models to detect unauthorized edits or malicious modifications.

Multimodal foundation models, such as diffusion models, are also susceptible to editing that may lead to malicious outputs. It is recommended that future discussions consider the unique challenges and potential defenses associated with protecting these models from tampering.

**Discussion Potential:**

3

**Paper Summary:**

Knowledge editing (KE) methods allow modifying facts in LLMs with minimal side effects, but they pose serious security risks. KE is widely accessible, efficient, and difficult to detect, making it attractive for malicious use.  Potential threats include misinformation, tampering, and unverified model updates. The lack of institutional awareness further exacerbates these risks. This paper urges the community to develop tamper-resistant models and enhance AI security.

**Position:**

Yes

**Position In Title:**

Yes

**Related Work:**

4

**Strengths And Weaknesses:**

Strengths:

The paper brings attention to an under-discussed area by highlighting how knowledge editing methods, despite their benefits, introduce significant security risks.

It effectively points out that these methods are not only efficient and cost-effective but also difficult to detect, making them attractive to malicious actors.

The discussion extends beyond technical aspects to include systemic vulnerabilities, such as unverified model updates and a lack of regulatory oversight.

By urging the community to develop tamper-resistant models and enhance overall AI security, the paper sets a proactive agenda for future research and policy-making.

**Support:**

4

---

> ### Author Rebuttal · Authors · 2025-03-31
>
> We thank the reviewer for the proposed extensions. We briefly comment on them below and will add a detailed discussion on these points in the CR, since we believe these will further enrich the paper.
>
> > Methods in the DPO series, such as “Knowledge Editing in Language Models via Adapted Direct Preference Optimization”, as well as model merging techniques, can also be exploited to implant attacks. It is recommended to supplement the discussion by considering how these approaches might be leveraged maliciously and to explore potential countermeasures.
>
> DPO: editing with KDPO might conceal a high risk, since the changes in weights caused by KDPO might be indistinguishable from those caused by DPO. This can make detecting KDPO-edits more challenging than other edits.
>
> Model merging techniques: merging several models, one of which is maliciously edited could result in a maliciously edited model as well. Counteracting such edits might require thoroughly examining the models before merging them.
>
> > Steering techniques can enable lightweight attacks since they are applied only once during the forward propagation—essentially resembling a user-initiated jailbreak. It is recommended to include those methods for discussion.
>
> Indeed, we fully agree that steering [1, 2, 3] can be taken as a form of model editing where the goal is to modify a specific behaviour, e.g., to avoid generating offensive languages rather than specific factual knowledge. In the paper, we focused on factual knowledge editing methods. We will incorporate a discussion on steering and merging techniques in the CR to provide a more holistic perspective.
>
> > A key question is whether a unified approach exists that can render the intended behavior non-editable, or if it is necessary to design distinct strategies for different methods. This question can help define a technical roadmap for future defenses. Moreover, one area worth exploring is whether interpretable techniques, such as SAE, can be used to "X-ray" large language models to detect unauthorized edits or malicious modifications.
>
> Making models non-editable may require the development of specialized optimization strategies that keep models at sharp minima. This approach ensures that any alterations to the model weights lead to a substantial degradation in the model's overall performance, thereby discouraging modifications. Nevertheless, despite the potential risks associated with knowledge editing, it can still be beneficial in certain scenarios, highlighting the need for a nuanced approach that balances model integrity with the potential benefits of editing  (please check our response to W1 from Reviewer s1y4 for details on the dual-use nature of model editing).
>
> Indeed, (mechanistic) interpretability tools are instrumental in analyzing model weights and uncovering the working mechanisms of LLMs. These tools facilitate in-depth analysis of LLMs, enabling the derivation of valuable insights into the behavior of both edited and unedited models.
>
> > Multimodal foundation models, such as diffusion models, are also susceptible to editing that may lead to malicious outputs. It is recommended that future discussions consider the unique challenges and potential defenses associated with protecting these models from tampering.
>
> Indeed, diffusion models can be maliciously edited to output harmful and biased content. Current work on countermeasures targets only LLMs, and developing countermeasures for multimodal foundation models represents a promising future direction.
>
> [1] https://arxiv.org/abs/2410.12877
>
> [2] https://arxiv.org/abs/2502.11356
>
> [3] https://arxiv.org/abs/2503.11314

---

### Official Review · Reviewer_uA4Z · 2025-03-13

**Significance:** 2
**Argument Clarity:** 3
**Rating:** 3
**Confidence:** 4

**Questions:**

Could you elaborate on how different knowledge editing methods (memory-based, meta-learning, and locate-and-edit) may present varying levels of security risks? Are certain approaches inherently more vulnerable to malicious exploitation than others?

In your security risk assessment, did you find any evidence that specific knowledge editing techniques are more resistant to misuse? If so, could this inform safer design practices?

Have you considered specific evaluation metrics to assess the effectiveness of the countermeasures you propose? What would a comparative evaluation framework look like?

What do you see as the most significant technical challenges in implementing your proposed safeguards, and how might these challenges be addressed in future work?

The paper highlights knowledge editing as a security risk, but similar techniques are also used in machine unlearning to enhance security by removing sensitive or harmful information. Could you clarify how the same methods can both pose risks and serve as safeguards? It would be helpful to see a discussion on this trade-off.

**Discussion Potential:**

3

**Paper Summary:**

This position paper contributes a comprehensive analysis of the overlooked safety risks posed by knowledge editing methods for Large Language Models. The authors discuss that KEs present serious security concerns through several arguments: (1) KEs possess properties (accessibility, affordability, performance, and stealthiness) making them attractive tools for malicious actors; (2) recent research has already demonstrated KEs' exploitation for harmful purposes including backdoor insertion, bias injection, jailbreaking, and misinformation spreading; (3) the AI ecosystem lacks verification mechanisms for model updates; and (4) there exists insufficient awareness at both social and institutional levels. The paper analyzes vulnerability across user groups and provides a discussion of current countermeasures' limitations, concluding with practical recommendations for future research directions including tamper-resistant models, verifiable updates, and conditional editing restrictions to improve model security.

**Position:**

Yes

**Position In Title:**

Yes

**Related Work:**

2

**Strengths And Weaknesses:**

Strengths:
1. The paper explains how knowledge editing methods work and why they're risky, backing this up with lots of real examples from recent papers.

2. The paper identifies specific groups that might be affected and suggests practical ideas for fixing the problems, like creating ways to detect edited models, developing methods to reverse malicious edits, and implementing model verification systems.

Weakness:
1. The paper treats all knowledge editing methods as equal security risks, but doesn't compare how different methods might vary in their vulnerability. The argument would be stronger if the paper explained which methods are more risky and under what conditions.

2. While the paper does discuss countermeasures, the proposed solutions lack technical depth compared to the threat analysis. The paper identifies several promising directions but provides limited details on implementation challenges, effectiveness metrics, or comparative evaluations of different countermeasure approaches.

3. The paper frames model editing purely as a security risk but overlooks its potential as a Machine Unlearning technique for removing harmful or outdated knowledge. A discussion on this trade-off, particularly in the Alternative Views section, could provide a more balanced perspective by considering whether KE could also serve as a controlled risk-mitigation tool rather than solely a threat.

**Support:**

2

---

> ### Author Rebuttal · Authors · 2025-03-31
>
> > W1: The paper treats all knowledge editing methods as equal security risks, but doesn't compare how different methods might vary in their vulnerability. The argument would be stronger if the paper explained which methods are more risky and under what conditions.
>
> We believe all knowledge editing methods can be used maliciously. However, some KEs might be easier to use than others, and the different KE approaches might necessitate implementing different countermeasures. Please refer to our answer to Q1/Q2 for more details.
>
> > W2: While the paper does discuss countermeasures, the proposed solutions lack technical depth compared to the threat analysis. The paper identifies several promising directions but provides limited details on implementation challenges, effectiveness metrics, or comparative evaluations of different countermeasure approaches.
>
>
> Please refer to our answer to Q3 below.
>
> > W3: The paper frames model editing purely as a security risk but overlooks its potential as a Machine Unlearning technique for removing harmful or outdated knowledge. A discussion on this trade-off, particularly in the Alternative Views section, could provide a more balanced perspective by considering whether KE could also serve as a controlled risk-mitigation tool rather than solely a threat.
>
> Please refer to our answer to Q5 below.
>
> > Q1: Could you elaborate on how different knowledge editing methods (memory-based, meta-learning, and locate-and-edit) may present varying levels of security risks? Are certain approaches inherently more vulnerable to malicious exploitation than others?
>
> > Q2: In your security risk assessment, did you find any evidence that specific knowledge editing techniques are more resistant to misuse? If so, could this inform safer design practices?
>
> We think both Q1 and Q2 go into the same direction and therefore provide a joint response. While all knowledge editing methods (KEs) can be used to malicious ends, some KEs might be easier to use than others (e.g., less data and compute requirements). Our analysis (cf. Table 2) shows that the computationally cheap KEs (e.g., ROME and In-context knowledge editing) are the most frequently used KEs for malicious uses. Additionally, the diverse KE approaches differ in the changes they cause in the model. Counteracting edits from different approaches might require detailed analysis of the model weights. KEs should ideally reveal both the fact that the model has been edited and the nature of edits. This guarantees high transparency towards users.
>
> > Q3: Have you considered specific evaluation metrics to assess the effectiveness of the countermeasures you propose? What would a comparative evaluation framework look like?
>
> Yes, depending on the countermeasures, evaluation metrics can be developed or be adopted from related work. For example, for edit detection, precision, recall and F1 are reported [1]. For reversing edits, matching accuracy, i.e., how often do the model’s outputs (after reversing) match the original model’s outputs [2]. We plan to provide more details on the proposed countermeasures in the CR, and sketch the major aspects for identifying whether a model has been edited here:
>
> * **Future work direction:** identifying edited models
> * **Evaluation:** accuracy in identifying edited models (overall and per editing method)
> * **Implementation challenges:** developing weight analysis tools to identify edited models
> * **Impact:** Identifying potential edits conducted with several methods
> * **Limitation:** no information on the type of the edits (malicious/benevolent) and the edits themselves
>
> > Q4: What do you see as the most significant technical challenges in implementing your proposed safeguards, and how might these challenges be addressed in future work?
>
> We believe that the diversity of knowledge editing approaches poses a significant challenge, as it necessitates the development of tailored countermeasures. For instance, both locate-and-edit KEs and meta-learning KEs modify model weights, but leave different traces in the model [1]. To effectively identify edited models resulting from these approaches, a thorough understanding of how each method impacts the model and the corresponding traces is essential. Therefore, we propose that future research focus on a detailed examination of these approaches and their effects. Leveraging existing mechanistic interpretability tools and developing novel ones could provide valuable insights and facilitate the development of more effective countermeasures.
> > Q5: The paper highlights knowledge editing as a security risk, but similar techniques are also used in machine unlearning to enhance security by removing sensitive or harmful information. Could you clarify how the same methods can both pose risks and serve as safeguards? It would be helpful to see a discussion on this trade-off.
>
> Please refer to our response to W1 from Reviewer s1y4.
>
> [1] https://arxiv.org/abs/2405.02765
>
> [2] https://arxiv.org/abs/2410.12586

---

### Official Review · Reviewer_s1y4 · 2025-03-14

**Significance:** 3
**Argument Clarity:** 3
**Rating:** 3
**Confidence:** 3

**Questions:**

- How can we balance the benefits of open-source LLMs with the need to ensure their security and prevent malicious modifications?
- How can we develop more robust methods for detecting and preventing malicious knowledge editing in LLMs?
- What role should AI developers, policymakers, and end-users play in addressing the risks of malicious knowledge editing?

**Discussion Potential:**

2

**Paper Summary:**

This paper discusses the security risks of knowledge editing (KE) in LLMs. The authors argue that KEs, which can modify specific facts in LLMs, pose a significant threat due to their accessibility, affordability, high performance, and stealthy nature. The paper also highlights the vulnerabilities in the LLM ecosystem that allow the unrestricted uploading and downloading of updated models without verification, and the lack of social and institutional awareness regarding these risks. The authors call for research into tamper-resistant models and countermeasures against malicious model editing.

**Position:**

Yes

**Position In Title:**

Yes

**Related Work:**

3

**Strengths And Weaknesses:**

**Strength:**
- The paper provides a thorough analysis of the risks associated with knowledge editing in LLMs, covering both technical and societal aspects.
- This paper provides detailed discussion of potential malicious use cases, such as backdoors and bias injection, effectively illustrates the dangers of KEs.
- It highlights the vulnerabilities in the AI ecosystem that enable the spread of maliciously edited models.

**Weakness:**
- The section on alternative views is somewhat brief and could be expanded to provide a more balanced discussion of the potential benefits and drawbacks of KEs.
- The paper discusses current countermeasures and their limitations but does not provide detailed proposals for new mitigation strategies, which could strengthen the call to action.
- While the paper presents a compelling argument, it lacks empirical evidence or case studies demonstrating real-world instances of malicious knowledge editing.

**Support:**

3

---

> ### Author Rebuttal · Authors · 2025-03-31
>
> > W1: The section on alternative views is somewhat brief and could be expanded to provide a more balanced discussion of the potential benefits and drawbacks of KEs.
>
> We  focused on the malicious uses of KEs to highlight the security risk they pose. We will add a thorough discussion on the benefits of KEs in the CR version to balance the view: Usage of KEs to remove private data from large language models [1], correct hallucinations [2], and defend against jailbreaking attacks [3]. Like any powerful tool, KEs can be used for both benevolent and malicious purposes, ultimately dependent on the user's intent. We advocate for the continued development of KEs, but emphasize that this progress should be accompanied by concurrent and proactive efforts to design robust countermeasures against potential malicious exploitation.
>
> > W2: The paper discusses current countermeasures and their limitations but does not provide detailed proposals for new mitigation strategies, which could strengthen the call to action.
>
> We do propose several future work directions in §6.2 (e.g., detecting edited models, inferring edited facts from model weights, reversing parameters-modifying edits, etc.) to mitigate the risk of malicious knowledge editing. These directions are based on limitations in related work, and our comprehensive analysis in the paper. We will rename section 6 to “Mitigation Strategies” and restructure it in the CR to make it more clear and add more details for the mitigation strategies like evaluation metrics and implementation challenges (please see our answer to Q3 of Reviewer uA4Z for further details) .
>
> > W3: While the paper presents a compelling argument, it lacks empirical evidence or case studies demonstrating real-world instances of malicious knowledge editing.
>
> We have referenced several related studies that demonstrate how knowledge editing can be used to implant backdoors, bias and jailbreak LLMs, and spread misinformation (cf. § 3.2, overview in Table 2). These studies investigate each malicious use case of knowledge editing in detail, and collectively support our position that knowledge editing poses serious safety risks.
>
> > Q1: How can we balance the benefits of open-source LLMs with the need to ensure their security and prevent malicious modifications?
>
> To mitigate potential risks in the current AI ecosystem, we propose to introduce safety mechanisms to verify the authenticity of uploaded models. Specifically, we recommend implementing checks to ensure that models have undergone the claimed update process, whether through fine-tuning, editing or both. Furthermore, users should be alerted when downloading models that are awaiting verification, allowing them to make informed decisions about the models they use. We shortly discuss this under §6.2 L428, and we will make it more clear in the CR version. Finally, in §3.4, we call for greater institutional awareness, which we view as essential to support the healthy and sustainable development of open-source LLMs.
>
>
> > Q2: How can we develop more robust methods for detecting and preventing malicious knowledge editing in LLMs?
>
> Given the proliferation of diverse knowledge editing (KE) approaches, including locate-and-edit, meta-learning, and memory-based methods, we believe it is essential to comprehensively analyze their effects on model weights and inference processes. Such an analysis would provide valuable insights, serving as a foundation for developing robust and targeted countermeasures to malicious knowledge editing. Developing countermeasures to malicious KE is the research we want to spark with this paper.
>
> > Q3: What role should AI developers, policymakers, and end-users play in addressing the risks of malicious knowledge editing?
>
> We believe that AI developers should raise awareness of the potential risks associated with open-source models and take these risks into account when integrating such models into their systems. Furthermore, we encourage researchers in AI safety to develop robust and effective countermeasures against malicious knowledge editing.  Policymakers should establish safety protocols that enhance the overall safety of the AI ecosystem. Furthermore, end-users should be educated about the potential for LLM outputs to be manipulated and factually incorrect, enabling them to critically evaluate the information they receive. We thank the reviewer for bringing up this point. We will include a more explicit discussion of it in the final version.
>
> [1] https://arxiv.org/abs/2406.18221
>
> [2]  https://arxiv.org/abs/2410.16251
>
> [3] https://arxiv.org/abs/2405.18166

---

### Official Review · Reviewer_ep9J · 2025-03-22

**Significance:** 2
**Argument Clarity:** 2
**Rating:** 2
**Confidence:** 5

**Questions:**

Q1: What are the specific new perspectives in your opinions on this topic?

Q2: I feel KE is not actually widely used in practice, FTs have even more risks and thus more important?

**Discussion Potential:**

3

**Paper Summary:**

This paper positioned that " Editing Large Language Models Poses Serious Safety Risks". The paper is centered around this topic by discussing the differences between KEs and other model updating strategies (e.g., finetuning and adapters), why is Knowledge Editing Risky, and additional potential impacts and discussions. In particular, the authors talked about many different aspects of knowledge editing, such as the Appealing Properties, Malicious Use Cases of Knowledge Editing, The Vulnerability of the AI Ecosystem and the Lack of social and institutional Awareness. The paper also provide some other Alternative Views on this topic.

I hold my opinion
## update after rebuttal

**Position:**

Yes

**Position In Title:**

Yes

**Related Work:**

3

**Strengths And Weaknesses:**

** Strengths**: this paper provides a very comprehensive overview of existing work on model editing and covers many topics about the potential benefits and risks of KE. It would serve as a good starting point for beginners to quickly learn the lastest progress in this area.

** Weaknesses**

I treat this paper more like a survey paper rather than a suitable position paper. First, most of those risks are already known and have even been discussed in many relevant works. Some survey papers [1,2] also provided a detailed discussion on this topic.

although the authors mentioned that "While this work is neither an exhaustive survey of PEFT methods", they still spent more than half page in table 1 to compare the differences between KE and PEFT, which is out of the main scope of this paper.


[1] https://arxiv.org/abs/2305.13172

[2] https://arxiv.org/abs/2401.01286

**Support:**

3

---

> ### Author Rebuttal · Authors · 2025-03-31
>
> > W1: I treat this paper more like a survey paper rather than a suitable position paper. First, most of those risks are already known and have even been discussed in many relevant works. Some survey papers [1,2] also provided a detailed discussion on this topic.
>
> We thank the reviewer for the two mentioned references [1,2]. While both of them provide valuable insights about knowledge editing, e.g., side effects of knowledge editing on the model performance, **neither of them discusses the risk of malicious knowledge editing**, which is the focus of this position paper. Rather than surveying model editing techniques as in the two referenced surveys, our work focuses on arguing why editing LLMs introduces serious safety risks.
>
>
> > W2: although the authors mentioned that "While this work is neither an exhaustive survey of PEFT methods", they still spent more than half page in table 1 to compare the differences between KE and PEFT, which is out of the main scope of this paper.
>
> Our intent was to highlight the specific desirable properties of Knowledge Editing (KE), like data-efficiency, compared to PEFT, which makes KE more appealing for malicious uses. We will emphasize this aspect more in the CR.
>
>
> > Q1: What are the specific new perspectives in your opinions on this topic?
>
> We take a holistic stance incorporating the desirable properties for malicious practitioners and potential malicious use cases alongside the vulnerability of the AI ecosystem, and the lack of social and institutional awareness. These points have not been explicitly discussed in prior work. By situating the discussion about malicious KE within this broader context, our paper aims to help the research community better recognize this risk and to motivate the development of appropriate countermeasures. To this end, we provide concrete suggestions that contribute to the shaping of these countermeasures.
>
>
> > Q2: I feel KE is not actually widely used in practice, FTs have even more risks and thus more important?
>
> While there is no information on how often knowledge editing methods (KEs) are used in practice, developing KEs continues to be an active area of research (https://researchtrend.ai/communities/KELM). Recognizing the potential dual-use nature of such technologies, in this paper we argue that the development of KEs must be accompanied by simultaneous and proactive efforts to design robust countermeasures against potential malicious exploitation of KEs.
>
> [1] https://arxiv.org/abs/2305.13172
>
> [2] https://arxiv.org/abs/2401.01286

---

### Decision · Program_Chairs · 2025-04-30

**Decision:**

Accept (poster)

**Comment:**

Most reviewers thought this paper has a clearly stated position, a good survey of relevant work in the area, comprehensive discussions on alternative reviews and a call to action to develop more temper resistant models. For weaknesses, reviewers would like to see more detailed proposals new mitigation strategies and perhaps a more balanced view on the positive uses of knowledge editing. The authors addressed most issues in the discussions.